# Volatile Profile of Portuguese Monofloral Honeys: Significance in Botanical Origin Determination

**DOI:** 10.3390/molecules26164970

**Published:** 2021-08-17

**Authors:** Alexandra M. Machado, Marília Antunes, Maria Graça Miguel, Miguel Vilas-Boas, Ana Cristina Figueiredo

**Affiliations:** 1Centro de Estudos do Ambiente e do Mar (CESAM Lisboa), Centro de Biotecnologia Vegetal (CBV), Faculdade de Ciências da Universidade de Lisboa, DBV, C2, Piso 1, Campo Grande, 1749-016 Lisboa, Portugal; ialexam@gmail.com; 2Centro de Estatística e Aplicações (CEAUL), Departamento de Estatística e Investigação Operacional, Faculdade de Ciências da Universidade de Lisboa, Campo Grande, 1749-016 Lisboa, Portugal; marilia.antunes@ciencias.ulisboa.pt; 3Faculdade de Ciências e Tecnologia, Mediterranean Institute for Agriculture, Environment and Development, Campus de Gambelas, Universidade do Algarve, 8005-139 Faro, Portugal; mgmiguel@ualg.pt; 4CIMO, Centro de Investigação de Montanha, Instituto Politécnico de Bragança, Campus de Santa Apolónia, 5300-253 Bragança, Portugal; mvboas@ipb.pt

**Keywords:** Portuguese honeys, volatile profile, honey type discrimination, botanical origin determination

## Abstract

The volatile profiles of 51 samples from 12 monofloral-labelled Portuguese honey types were assessed. Honeys of bell heather, carob tree, chestnut, eucalyptus, incense, lavender, orange, rape, raspberry, rosemary, sunflower and strawberry tree were collected from several regions from mainland Portugal and from the Azores Islands. When available, the corresponding flower volatiles were comparatively evaluated. Honey volatiles were isolated using two different extraction methods, solid-phase microextraction (SPME) and hydrodistillation (HD), with HD proving to be more effective in the number of volatiles extracted. Agglomerative cluster analysis of honey HD volatiles evidenced two main clusters, one of which had nine sub-clusters. Components grouped by biosynthetic pathway defined alkanes and fatty acids as dominant, namely *n*-nonadecane, *n*-heneicosane, *n*-tricosane and *n*-pentacosane and palmitic, linoleic and oleic acids. Oxygen-containing monoterpenes, such as *cis*- and *trans*-linalool oxide (furanoid), hotrienol and the apocarotenoid α-isophorone, were also present in lower amounts. Aromatic amino acid derivatives were also identified, namely benzene acetaldehyde and 3,4,5-trimethylphenol. Fully grown classification tree analysis allowed the identification of the most relevant volatiles for discriminating the different honey types. Twelve volatile compounds were enough to fully discriminate eleven honey types (92%) according to the botanical origin.

## 1. Introduction

Honey is a food product that has been consumed by mankind for since at least 6000 years [1]. Carbohydrates are the main components, comprising about 95% of honey’s dry weight, including the dominant monosaccharides fructose and glucose, in addition to about 25 different oligosaccharides [2]. Other constituents, such as water (10–20% *w*/*w*), the second-largest component of honey, and other minor compounds are present, such as proteins, free amino acids, enzymes, organic acids, lipids, volatiles, vitamins, minerals and phenolic compounds [3,4]. Honey consumption has increased in the last few years mainly due to a growth in the world population as well as due to a preference for natural foods of a rising number of consumers [5].

According to both Codex Alimentarius and the European Honey Directive [6,7], honey must not have any added food ingredients, including food additives or other substances, apart from other forms of honey. The authenticity of this food product comprises two main aspects, the origin of the honey, which includes botanical source and geographical origin, and the production method related to the harvesting of honey and its processing [8]. Honey’s botanical source strongly affects its organoleptic properties [9,10], monofloral honey being more in demand compared to multifloral, due to its biological activities as well as the characteristic aroma and taste [11]. Honey is recognised as a high-quality food product, its falsification being a major problem around the world since it is one of the most adulterated products, after olive oil and milk [5,10,12,13,14]. The increase in the number of consumers concerned about honey’s authenticity and environmental sustainability [15] has increased the demand for reliable analytical methods to establish criteria to guarantee honey authenticity [10].

Honey’s aroma depends on the volatile compounds present, mostly in the nectar, which help to discriminate honey types from different botanical and geographical origins [16,17,18,19,20]. The mainland Portugal and Azores and Madeira Islands are characterised by a rich and varied honey flora, contributing to the production of a great diversity of local monofloral honeys. Most representative monofloral honeys include those obtained from the mainland from bell heather (*Erica* spp.), eucalyptus (*Eucalyptus* spp.) and lavender (*Lavandula luisieri*, *L. pedunculata*, *L. stoechas*). Other monofloral honeys are also important, such as those from chestnut (*Castanea sativa*), incense (*Pittosporum undulatum*) from Azores Islands, orange (*Citrus* spp.), pennyroyal (*Mentha pulegium*), rosemary (*Rosmarinus officinalis*), strawberry tree (*Arbutus unedo*), sunflower (*Helianthus annuus*) and viper’s bugloss (*Echium* spp.) [21]. In 2019, honey production in Portugal was about 10,104 t, including 127 t and 46 t from Azores and Madeira Islands, respectively [22]. Only a small number of studies concern the characterisation of volatiles of Portuguese honeys, namely bell heather, chestnut, eucalyptus and multifloral honeys [23,24,25,26].

Whereas gas chromatography–mass spectrometry (GC-MS) is the technique mostly used to identify volatiles, different volatile extraction techniques have been used, such as hydrodistillation (HD), solid-phase microextraction (SPME), liquid–liquid extraction, simultaneous distillation extraction, simultaneous steam distillation–dichloromethane extraction (using Likens–Nickerson apparatus), ultrasonic solvent extraction or the purge-and-trap technique [19,27,28].

With a view to integrate knowledge on the volatiles of Portuguese monofloral honeys and determine whether they can be used as a tool for honey authenticity determination, this work aimed at defining (a) the volatile composition of Portuguese monofloral honeys from different botanical origins by two different methods: SPME and HD; (b) whether there is a correspondence between the flower’s volatiles and the monofloral honey type’s volatiles; (c) the degree of correlation between the honey samples; and (d) the discriminatory potential of specific volatile compounds regarding the botanical origins of different monofloral honey types.

## 2. Results and Discussion

### 2.1. Honey and Flower Volatile Profiles

The volatiles from 12 monofloral-labelled Portuguese honey types were assessed in a total of 51 samples (Figure 1, Table 1, Appendix A). Honey volatiles were isolated by two different methods, solid-phase microextraction (SPME) and hydrodistillation (HD) (Figure 1). To evaluate whether the presence of some honey volatiles is due to the volatiles from flowers visited by the bees, the main components obtained by HD from some selected flower species were also determined (Table 1, Appendix A).

#### 2.1.1. Volatiles Collected by Solid-Phase Microextraction (SPME)

The relative amounts of all the identified compounds obtained by SPME are listed in SM Appendix A according to the minimum and maximum percentage range of components found for honey samples. In total, 61 volatiles were identified in the 51 analysed honey samples. The identified compounds were dominated by benzoic acid derivatives and oxygen-containing monoterpenes, ranging from not detected (nd) to 59% and from nd to 47%, respectively. Alkanes and fatty acid esters were also identified in most of the samples. Apocarotenoids were found in great amounts in a specific group comprising strawberry tree honeys, ranging from 60 to 88%. Since with SPME, a lower number of volatiles was detected and their vast majority was also detected with HD isolation procedure, HD data were selected for further processing.

#### 2.1.2. Volatiles Collected by Hydrodistillation (HD)

The relative amounts of all the identified volatile compounds isolated by HD from the 51 monofloral-labelled honey samples are listed in SM Appendix A in order of their elution on the DB-1 column. One hundred ninety-two (192) volatile compounds were identified in the analysed honey samples. The main identified compounds included hydrocarbons, mostly alkanes (11 to 77%). Alkenes (nd–24%), methyl-branched alkanes (nd–2%), long-chain fatty acids (1–45%), their esters and aliphatic fatty alcohols (1–16%) were also found in relatively high percentages. In lower amounts, oxygen-containing monoterpenes and sesquiterpenes were identified in all honey samples, ranging from <0.05 to 27% and from <0.05 to 4%, respectively. Aromatic amino acid derivatives were also found in all samples (0.1–25%), as well as green leaf volatiles (GLVs) in lower amounts, although GLVs were not identified in the strawberry tree honey samples. Apocarotenoids were the dominant group found in this honey type (5–44%).

The analysis of selected flower volatiles confirmed the presence of common compounds identified both in the flowers as well as in the corresponding monofloral honeys (Table 1).

The volatile profile obtained by the hydrodistillation of each of the honey types analysed is, in this section, compared with data on the honey’s volatile marker compounds from literature.

Carob tree honey samples showed predominance of *cis*- and *trans*-linalool oxides (furanoid and pyranoid), (0.1–15% and <0.05–5% correspond to the furanoid form, *-cis* and -*trans*, respectively, and (nd–0.3%, nd–0.6%) correspond to the pyranoid form, -*cis* and -*trans*, respectively, also identified in the volatile profile of carob tree flowers. These compounds were also identified in the flower volatiles from *C. siliqua* collected in the south of Portugal [29]. *cis*- and *trans*-Linalool oxides (furanoid) and hotrienol have also been identified as the main marker volatile compounds in carob tree honey samples from Morocco and Balearic Islands [30,31]. In the current work, *cis*- and *trans*-linalool oxides (pyranoid) and *trans*-β-damascenone (nd–0.5%) were also considered as possible marker compounds for this honey type.

2′-Aminoacetophenone (<0.05–0.7%) as well as acetophenone (0.1–0.4%) were the characteristic volatiles identified in chestnut honey samples, with the second being also detected in the corresponding flowers. Chestnut honey aroma has been characterised by the presence of 2′-aminoacetophenone, acetophenone, benzaldehyde, phenyl ethyl alcohol, linalool, *n*-nonanol and *n*-nonanal, as reported in Croatian, Italian and Spanish chestnut honeys [32,33,34]. A previous work with Portuguese chestnut honey [25] showed benzaldehyde, *n*-nonanol, *n*-nonanal and linalool in common with the current study. The results obtained herein indicate acetophenone and 2′-aminoacetophenone as volatile markers for this honey type, with only acetophenone identified in the flowers.

Eucalyptus honey samples evidenced 1-nonanol (<0.05–0.6%), α-eudesmol (0.1–0.5%), β-eudesmol (0.1–0.4%), *n*-nonanal (0.1–0.4%), aromadendrene (0.0–0.4%), 2-undecanol (nd–0.3%) and *p*-cymen-8-ol (nd–0.2%) as characteristic compounds, with α- and β-eudesmol as well as aromadendrene also present in the respective flowers. 4-Keto-isophorone, 2-hydroxyisophorone, nerol oxide, *p*-cymene and *p*-cymen-8-ol contribute to the recognisable eucalyptus honey aroma [35,36,37], in agreement with the current results. The correspondence found with the sesquiterpene aromadendrene between eucalyptus flowers and the honey samples could indicate this compound as a possible volatile marker for Portuguese eucalyptus honey.

The distinguishing volatiles found for bell heather honey were benzene acetaldehyde (2–24%), *cis*-linalool oxide (furanoid, 2–12%), hotrienol (2–10%), *trans*-linalool oxide (furanoid, 1.0–5%), 1,2-dihydro-1,1,6-trimethyl-naphtalene (<0.05–3%), benzaldehyde (0.1–2%) and benzylalcohol (<0.05–0.9%). 1,2-Dihydro-1,1,6-trimethyl-naphtalene was likewise identified in bell heather flowers, with benzaldehyde and benzene acetaldehyde. Samples H1 and H3 were characterised by having benzene acetaldehyde (17–24%) and hotrienol (3–7%) as dominant compounds, while samples H4 and H6 exhibited *cis*-linalool oxide (furanoid, 9–12%), hotrienol (3–10%) and *trans*-linalool oxide (furanoid, 4–5%) as the main compounds. Samples H2 and H5 showed lower percentages of these compounds. Benzene acetaldehyde, *cis*-linalool oxide and hotrienol have also been mentioned as characteristic compounds in studies with Portuguese and Spanish bell heather honeys [24,25]. Although benzene acetaldehyde could be a Strecker degradation product of phenylalanine [28] or used as a bee repellent to facilitate honey collection [38], this compound was also identified in other studies with different honey types [39]. Phenylalanine is one main amino acid present in bell heather honey [40], and so its content in the correspondent plants should be determined to obtain a relationship between it and benzene acetaldehyde. Compounds shared with other honey types, such as isovaleric acid, 2-methylbutyric acid, benzyl alcohol and phenyl ethyl alcohol were also identified. Other studies have also reported the presence of apocarotenoids, such as α-isophorone [41] and decanoic acid and *p*-anisaldehyde, also identified in the current work, indicating a botanical origin within the Ericaceae family [42].

Benzyl salicylate was the distinguishing compound of incense honey (nd–0.3%), also detected in the corresponding flowers. This could be indicative of a marker compound for this honey type. However, other volatiles were also found, namely benzene acetaldehyde as well as *cis*-linalool oxide (furanoid) in the same amounts (1–3%), *trans*-linalool oxide (furanoid, 0.6–2%), benzaldehyde (<0.05–0.4%), α-eudesmol (nd–0.4%), α-terpineol (<0.05–0.3%) and limonene (nd–0.3%). Incense honey samples were characterised by the absence of hotrienol and by the presence of oleic acid as the main component (17–39%). There are few studies regarding this honey type, this being the first time that its volatile compounds were identified.

Lavender honey samples showed *n*-nonanal (nd–0.8%) and *n*-decanal (nd–0.4%) as differentiating compounds, although they were not found in the analysed flowers. Benzaldehyde, hexanoic acid, benzene acetaldehyde and phenyl ethyl alcohol were also detected, as mentioned in another work with Portuguese lavender honey specifically obtained from *Lavandula stoechas* [43]. The volatiles’ composition suggests that lavender honey is distinctive relative to the other analysed honey types; however, no specific markers were found.

Several distinctive volatile compounds were identified in orange honey, such as methyl anthranilate (nd–2%); lilac aldehydes A (nd–0.8%), B (nd–1%), C (nd–0.7%), D (nd–0.2%) and E (nd–0.05%); indole (nd–0.3%) and *cis*-myrcenol (nd–0.2%). With the exception of lilac aldehydes D and E, the remaining compounds were also identified in orange flowers, which could confirm their botanical origin. Lilac aldehydes A, B and C and methyl anthranilate have also been mentioned by other authors as markers for orange honeys from Greece and Spain [44,45,46]. Additionally, indole was identified in this study in both flowers and honey, which may indicate that this compound has a possible marker for *Citrus* spp. honey type, and has also been identified by other authors in the nectar of orange blossoms [39,47].

Rape honey samples only evidenced dimethyl trisulphide as a distinguishing compound in trace amounts (<0.05%), which is in agreement with the results reported by other works [48,49]. These samples were relatively poor in the number of volatiles, and they were common to several honeys, such as benzene acetaldehyde and hotrienol, also reported for rape honeys obtained from the Czech Republic [50].

Raspberry and rosemary honey samples did not evidence specific volatile compounds. There are only a few studies on raspberry honey volatiles and report compounds in common with the present work, for instance, *cis*- and *trans*-linalool oxide (furanoid); hotrienol; and lilac aldehydes A, B and C identified in Slovakian honey [51].

The alcohols 3-methyl-1-butanol and 3-methyl-3-buten-1-ol, the apocarotenoid 4-keto-isophorone and lilac aldehyde isomers have been previously identified in rosemary honeys [35,52]. Whereas these compounds were not detected in the present study, others were, which are considered ubiquitous honey constituents, such as benzaldehyde, benzene acetaldehyde and hotrienol. Probably because only one sample was studied in this work, no marker compounds were found for this honey type.

The main volatiles of sunflower honey’s were α-pinene (<0.05–0.2%), also detected in large amounts in the flowers, as well as in other honey types and flower volatiles. Benzene acetaldehyde (0.1–1%) and β-copaene (0.1–0.2%) were also detected in lower amounts. α-Pinene was the only compound identified in this honey, in common with the honeys of other countries, namely the Czech Republic, Romania and Spain [53]. However, β-copaene was only identified in this honey type, suggesting it as a possible marker for Portuguese sunflower honey.

Strawberry tree honey samples evidenced the presence of the following volatiles: α-isophorone (5–39%), 3,4,5-trimethylphenol (2–13%), 2,3,5-trimethylphenol (0.4–4%), 4-keto-isophorone (0.4–3%), 2-hydroxyisophorone (0.2–3%), veratrole (nd–2%), 1,2-dihydro-1,1,6-trimethyl-naphtalene (<0.05–1%), edulan (<0.05–0.7%), 5-methyl-3-hexen-2-one (<0.05–0.4%) and 2,6-dimethylpyrazine (nd–0.1%). In common between the flowers and the honey were α-isophorone, 2,3,5-trimethylphenol, 3,4,5-trimethylphenol and 4-keto-isophorone, suggesting them as possible markers for strawberry tree honey. The apocarotenoids identified, α-isophorone and 4-oxoisophorone, have also been reported by other authors as floral markers for this honey type produced in Greece and Sardinia [54,55,56]. Furthermore, in this study, other compounds were also identified, such as the aromatic amino acid derivatives veratrole and 2,3,5-trimethylphenol, as well as 3,4,5-trimethylphenol in higher amounts. Despite not being identified in the present work, benzene derivatives (dimethyl, trimethyl, tetramethyl) were identified by Karabagias et al. [54] in Greek strawberry tree honeys.

Notwithstanding their contribution to the honey volatile profile, some identified compounds cannot be considered as reliable botanical origin honey markers, not only due to their presence in the comb as well as in bees and cuticular wax, but also because of their general occurrence in different honey types. Among these are alkanes (*n*-tricosane, *n*-pentacosane, *n*-heptacosane), alkenes ((*Z*)-9-tricosene and (*Z*)-9-pentacosene)), methyl-branched hydrocarbons, long-chain fatty acids (oleic acid) and their esters (ethyl oleate), as well as aliphatic fatty alcohols (1-hexadecanol, oleyl alcohol, 1-octadecanol). The results reported herein are in line with those of different authors [57,58,59,60].

Among the compounds common to different honey types that cannot be considered as specific volatile markers are benzaldehyde, benzene acetaldehyde, benzyl alcohol, isovaleric acid, 2-methylbutyric acid and linalool derivatives, such as *cis*- and *trans*-linalool oxides (furan isomers). Compounds that could be generated in honey during heating were also identified, namely furan and pyran derivatives [28]. However, the amount of these compounds, which could be identified as artefacts, was quite small (≤2.3%), since the volatile extraction was performed for just 1 h to diminish the occurrence of hydrolysis reactions and the formation of sugar degradation products [61].

Despite several studies describing the identification of specific volatile compound markers in some honey types [30,35,44,62,63], only a few compounds seem really specific, with variable concentrations according to the different plant sources.

### 2.2. Statistical Analysis

Since HD was more effective in the number of isolated volatiles, statistical analysis was performed with the results obtained by this extraction method using all 51 honey samples.

#### 2.2.1. Cluster Analysis

Hierarchical clustering was used to find whether the samples of each monofloral honey type were chemically correlated and whether there was correlation between honey types based on the composition of the volatiles (Figure 2, Table 1 and Table 2). Cluster analysis evidenced a dendrogram with two main clusters, one of which had nine sub-clusters. Despite the main clusters, I and II, showing very low correlation (Scorr < 0.12), in general, honeys from the same botanical origin were assigned to the same cluster and sub-cluster, highlighting their volatiles’ similarity.

Cluster I included nine moderately correlated sub-clusters, Ia–Ii (Scorr < 0.44), in which were included all honey types except three samples of strawberry tree honey (cluster II). Cluster I was characterised by the dominance of straight-chain hydrocarbons *n*-C23 (6–29%), *n*-C9 (1–26%), *n*-C21 (1–26%), *n*-C5 (0.3–16%) and fatty acids, as well as their derivatives oleic acid (nd–39%), ethyl oleate (nd–11%) and linoleic acid (nd–9%). Oxygen-containing monoterpenes such as *cis*- and *trans*-linalool oxide (furanoid) were also present, ranging from nd to 15% and nd to 5%, respectively.

Four carob tree honey samples evidenced high correlation (Scorr < 0.82) in sub-cluster Ia (Ct1, Ct2, Ct3, Ct5), while Ct4, positioned in sub-cluster Ig, showed moderate correlation with the former ones (Scorr < 0.58). Both chestnut (Scorr < 0.86) and eucalyptus honey samples were highly correlated in sub-cluster Ia (Scorr < 0.82).

Bell heather honey samples evidenced different main volatiles according to geographical region, highlighting its influence. The six bell heather honey samples formed three groups with two samples in each. Specifically, sub-cluster Ia showed very highly correlated (Scorr < 0.92) H2 and H5, both obtained from Trás-os-Montes e Alto Douro; sub-cluster Ig had highly correlated (Scorr < 0.80) H4 and H6 from the Minho region; and sub-cluster Ii had highly correlated (Scorr < 0.90) H1 and H3 from Beira Baixa. Benzene acetaldehyde was dominant in the Ii cluster.

Sub-cluster Ih comprised the four incense honey samples I1-I4, highly correlated (Scorr < 0.90), evidencing oleic acid (17–39%) and linoleic acid (nd–9%) as the main components.

Lavender honey samples were all highly correlated (Scorr < 0.72), with six of them in sub-cluster Ia (L1, L2, L4–L6, L8), while samples L3 and L7 were isolated in sub-clusters Ib and Ic, respectively.

Orange honey samples were positioned in three separate sub-clusters. Ia had the highly correlated (Scorr < 0.86) O2 to O6 and O9 samples; sub-cluster If had very highly correlated (Scorr < 0.98) O7 and O8; and O1 was placed alone in sub-cluster Ih.

Rape honey samples were positioned in different sub-clusters, R1 in Ie and R2 in Ia, showing moderate correlation between them (Scorr < 0.68).

Raspberry honey samples also exhibited moderate correlation (Scorr < 0.52), with Rb1 placed in sub-cluster Ih and Rb2 in Ia.

The only rosemary honey sample (Ro1) was found in sub-cluster Ia.

Sf1 and Sf3 sunflower honey samples were very highly correlated (Scorr < 0.94) in sub-cluster Ia, and sample Sf2 was placed in sub-cluster Id, moderately correlated with the other two (Scorr < 0.70). Sunflower honey samples produced in Alentejo remained together in the dendrogram, while the other, obtained from Algarve, was positioned in a different sub-cluster, indicating an influence of the geographical origin on the volatile composition. The compounds β-myrcene, benzene acetaldehyde, 2,3,5-trimethylphenol, palmitic acid and oleyl alcohol could be geographical region dependent for this honey type.

Cluster II, with three samples of strawberry tree honey, showed α-isophorone (36–39%) and 3,4,5-trimethylphenol (4–13%) as the main compounds. The three samples included (St1–St3) showed very high correlation (Scorr < 0.96), while the other strawberry tree honey sample, St4, placed in sub-cluster Ia, showed very low correlation with the former ones (Scorr < 0.12). The differences observed could be explained due to geographical reasons. Three of them were obtained from Alentejo and Algarve (St1–St3), while St4, with a lower amount of apocarotenoids and placed in a different cluster, was also produced in Algarve but in a region with peculiar geographical and climatic conditions, in Aljezur, surrounded by the sea and the hills, reflecting this dual influence also on honey volatile composition.

#### 2.2.2. Classification Tree

A fully grown classification tree was built with the volatiles obtained by each honey’s hydrodistillation (Figure 3). It was found that 12 compounds were enough to fully discriminate 11 out of the 12 honey types, detailed in the bottom nodes, according to the botanical origin. The discriminant compounds were benzaldehyde, benzene acetaldehyde, β-copaene, *cis*-linalool oxide (furanoid), *n*-decane, ethyl hexadecanoate, α-eudesmol, 2-furfural, heptacosene, hotrienol, *n*-tricosane and UI 8. The square at each node shows the honey types present in the node, the number of honey samples included and the percentage of the data encompassed in that node. At the root (top node), the number of labelled honey samples for each honey type is shown in two rows. The first row (from left to right) exhibits two of the chestnut honeys, five of the carob tree honeys, five of the eucalyptus honeys, six of the bell heather honeys, four of the incense honeys and eight of the lavender honeys. From left to right, the second row shows the remaining samples according to honey types: nine of orange, two of rape, two of raspberry, one of rosemary, three of sunflower and four of strawberry tree. The root node contains 100% of the data. The first split (evaluation of the condition UI 8 < 0.25) puts 82% of the data in a group where the lavender honey is the most frequent honey type and 18% in a group where the orange honey is the most frequent. After this split, the classification tree continued the discrimination of the honey types, according to volatile compounds, along the internal nodes, and at the bottom of the tree, most of the samples were included in the node labelled with the correspondent honey type.

The classification tree suggests that the 12 identified volatile compounds are potentially good discriminators of honey types, since they were enough to fully discriminate 11 different monofloral honeys under analysis. These are important results because they have allowed pinpointing, from a large number of volatiles, some compounds able to discriminate honey types. Hotrienol, identified as one of the dominant compounds in heather honey, was used to discriminate this honey type, while *cis*-linalool oxide (furanoid) was used for carob tree and α-eudesmol for eucalyptus honey samples. β-Copaene, only identified in sunflower honey samples, was also useful to discriminate this honey type.

In the present global market, food product authenticity allied to quality assurance is a mandatory requirement. Honey, as a food product, linked with the growing use in the pharmaceutical industry, is widely consumed all over the world. The guarantee of honey authenticity can be achieved through a joint effort, with the integration of several analysis techniques. Analysis of honey’s volatiles is one additional technique that can be applied to the study of monofloral honeys to distinguish between those from different botanical sources as well as geographical origins. This work showed that is possible to differentiate monofloral honeys according to the identified volatile compounds and, in some cases, geographical locations.

The results of this study increased our knowledge about the dominant compounds found in different honey types produced in several regions in mainland Portugal and the Azores Islands. The detection of some specific compounds in the honeys, also identified in the respective flowers, allowed understanding the origin of marker compounds, characteristic of the analysed monofloral honeys. Further work on honey volatiles with a larger number of samples per honey type would help build a robust classification tool, allowing discrimination between honeys from different botanical sources. This knowledge could be useful for other studies on the nutritional and therapeutic potential of Portuguese honey.

## 3. Materials and Methods

### 3.1. Honey Sampling

Fifty-one honey samples labelled as monofloral according to botanical origin were collected from several regions of mainland Portugal, specifically Minho, Trás-os-Montes e Alto Douro, Beira Baixa, Estremadura, Alto Alentejo, Baixo Alentejo, Alentejo Litoral and Algarve, and from Azores Islands, São Miguel and Pico, as detailed in Table 1. Most of the samples were obtained from producers and the remaining from specialised shops, between 2015 and 2018. Honey samples were labelled with 12 different botanical origins when acquired. The honey samples were stored in a cool, dry place until further assessment. All the samples were subjected to pollen analysis to confirm their botanical origin, as previously described [64]. Flowers corresponding to the different honey types were collected from different geographical locations in Portugal, as detailed in Table 1.

### 3.2. Honey and Flower Volatile Sampling

The volatiles from 51 monofloral honeys were isolated by two different methodologies, (a) solid-phase microextraction (SPME) and (b) hydrodistillation, to compare the efficiency of volatile extraction. Flower volatiles were isolated by hydrodistillation.

#### 3.2.1. Sampling of Honey Volatiles by SPME

Headspace volatiles were collected by SPME using a 100 μM polydimethylsiloxane (PDMS)-coated fibre (Supelco, Bellefonte, PA, USA) inserted into a manually operated SPME holder. Each SPME fibre was thermally conditioned for up to 20 min at 250 °C according to the manufacturer’s recommendations before use. Blank assays of the fibres were performed regularly.

For volatiles’ collection and after lid removal, each of the monofloral-labelled honey flask was inserted into a glass desiccator (ø 20 cm) (Figure 1, top row) and left to stand for 1 h at room temperature for atmosphere homogenisation. The conditioned fibres were exposed in the desiccator for 1 h. Three SPME fibres were used per desiccator, with two being analysed by gas chromatography (GC) with flame ionisation detection for quantification of volatiles and the other being analysed by gas chromatography–mass spectrometry (GC-MS) for chemical composition analysis. Preliminary experiments were run to ascertain the optimal period and temperature of fibre exposure.

#### 3.2.2. Sampling of Honey and Flower Volatiles by Hydrodistillation (HD)

Volatiles were obtained by hydrodistillation using a Clevenger-type apparatus according to the European Pharmacopoeia (Council of Europe, 2010). In the case of honeys, hydrodistillation was run for 1 h (Figure 1, bottom row) and from 1 to 3 h for flowers, depending on the amount of the plant material available (Appendix A). Approximately 50 g of each honey, used in SPME analysis, was added to 100 mL of distilled water before hydrodistillation. The extracted organic compounds were recovered with in-lab distilled pentane from the supernatant layer on the distilled water (hydrolate) that remained after the hydrodistillation was complete. The pentane solution was then concentrated to a volume of ~100 µL at room temperature under a stream of nitrogen. Honey and flower volatiles were stored at −20 °C until analysis.

### 3.3. Analysis and Quantification of Compounds

Volatiles were analysed by GC for quantification and by GC-MS for component identification.

#### 3.3.1. Gas Chromatography (GC)

**SPME samples.** Immediately after sampling, the SPME needle was introduced into the split/splitless injector of a PerkinElmer Clarus 400 gas chromatograph (PerkinElmer, Waltham, MA, USA) equipped with two flame ionisation detectors with a data handling system. Two columns of different polarities were inserted into the injector port: a DB-1 fused-silica column (100% dimethylpolysiloxane, 30 m × 0.25 mm i.d., film thickness 0.25 μm; J & W Scientific Inc., Folsom, CA, USA) and a DB-17HT fused-silica column ((50–phenyl)-methylpolysiloxane, 30 m × 0.25 mm i.d., film thickness 0.15 μm; J & W Scientific). The oven temperature was programmed to rise from 45 to 175 °C at 3 °C/min, then to 300 °C at 15 °C/min and finally to remain isothermal for 10 min, for a total run time of 61.67 min. The SPME fibre was desorbed in splitless mode for 1 min, and the injector and detector temperatures were 250 and 290 °C, respectively; the carrier gas was hydrogen, adjusted to a linear velocity of 30 cm/s. The percentage composition of the volatiles was computed by the normalisation method from the GC peak areas, without the use of correction factors, calculated as mean values of two SPME fibres from each sample.

**Hydrodistillation samples.** Volatiles obtained from honey and flowers were analysed using the same equipment and settings as described in this section for SPME samples, with the following exceptions: the split injector ratio was 1:40, and the injector temperature was 280 °C. The percentage composition of the volatiles was computed by the normalisation method from the GC peak areas, without the use of correction factors, calculated as mean values of two injections from each sample.

#### 3.3.2. Gas Chromatography–Mass Spectrometry (GC-MS)

**SPME samples.** Directly after sampling, the SPME needle was introduced into the split/splitless injector of a PerkinElmer Clarus 600 gas chromatograph equipped with a DB-1 fused-silica column (100% dimethylpolysiloxane, 30 m × 0.25 mm i.d., film thickness 0.25 μm; J & W Scientific), interfaced with a PerkinElmer Clarus 600T mass spectrometer (software version 5.4.2.1617, PerkinElmer, Shelton, CT, USA). Injector and oven temperatures were as indicated in Section 3.3.1; the transfer line temperature was 280 °C; the ion source temperature was 220 °C; the carrier gas was helium, adjusted to a linear velocity of 30 cm/s; analyte desorption was achieved in splitless mode for 1 min; the ionisation energy was 70 eV; the scan range was 40–300 u; and the scan time was 1 s.

The identities of the components were assigned by a comparison of their retention indices (RIs) to C_6_–C_31_ *n*-alkane (Sigma) indices and GC-MS spectra from a laboratory-made library based upon the analyses of reference essential oils, laboratory-synthesised components and commercially available standards.

**Hydrodistillation samples.** The equipment and the analysis settings were as above, with the exceptions already mentioned in Section 3.3.2. In addition, the apocarotenoid edulan could be identified by comparing with one of the authentic samples present in the reference purple passion fruit [65].

### 3.4. Statistical Analysis

#### 3.4.1. Cluster Analysis

The percentage composition of the isolated volatiles was used to determine the relationship between the different samples by cluster analysis using the Numerical Taxonomy Multivariate Analysis System (NTSYS PC software, version 2.2, Exeter Software, Exeter University, Exeter, UK). For cluster analysis, the correlation coefficient was selected as a measure of similarity among samples and the unweighted pair group method with arithmetical averages (UPGMA) was used for cluster definition. The degree of correlation was evaluated according to [66] as very high [0.90, 1.00], high [0.70, 0.90], moderate [0.40, 0.70], low [0.20, 0.40] and very low (<0.20).

#### 3.4.2. Classification Tree

Considering the botanical sources of honey, a classification tree was built using the volatile components obtained by hydrodistillation. The tree was fully grown with the purpose of investigating whether a small number of volatiles would be enough to nearly completely discriminate the honey types. This approach had a discovery purpose rather than the aim of building a classification tool. The analysis was performed in R (version 4.0.2) and RStudio (version 1.3.1093) using the package rpart [67].

## Figures and Tables

**Figure 1 molecules-26-04970-f001:**
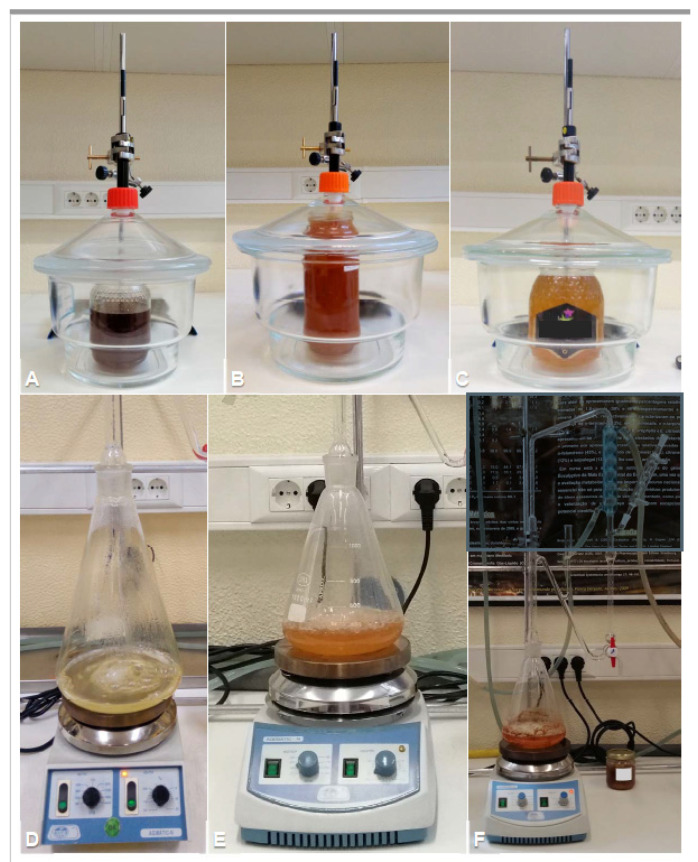
Extraction of volatiles from different honey samples. (**A**–**C**) SPME collection of volatiles from carob tree honey (**A**), eucalyptus honey (**B**) and lavender honey (**C**). (**D**–**F**) Hydrodistillation of orange honey (**D**), sunflower honey (**E**) and strawberry tree honey (**F**).

**Figure 2 molecules-26-04970-f002:**
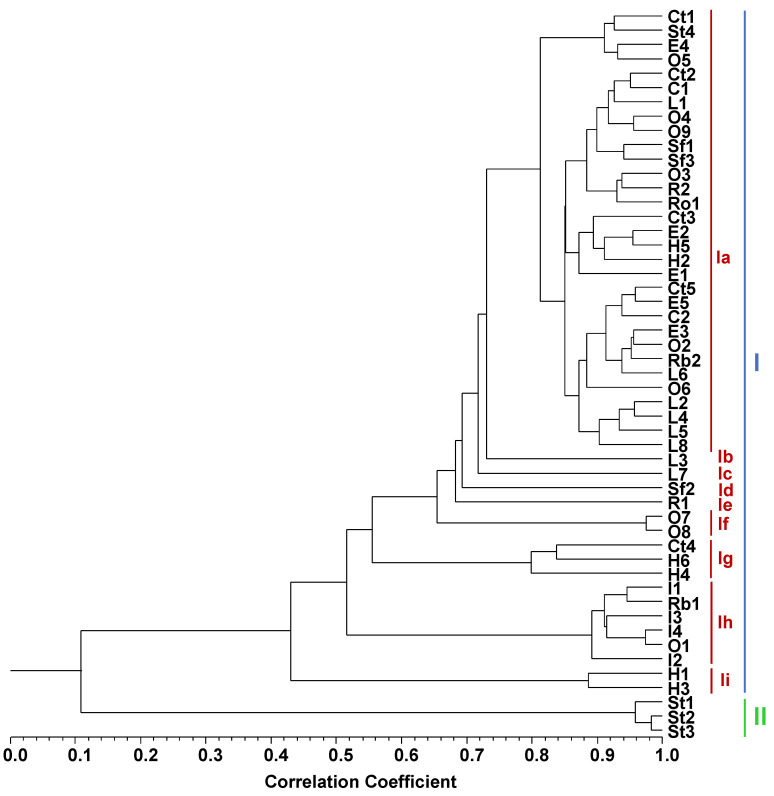
Dendrogram obtained by cluster analysis of the percentage composition of volatile compounds isolated by hydrodistillation from the 51 honey samples, based on correlation, and using the unweighted pair-group method with arithmetic average (UPGMA). Ct: carob tree; C: chestnut; E: eucalyptus; H: bell heather; I: incense; L: lavender; O: orange; R: rape; Rb: raspberry; Ro: rosemary; Sf: sunflower; St: strawberry tree.

**Figure 3 molecules-26-04970-f003:**
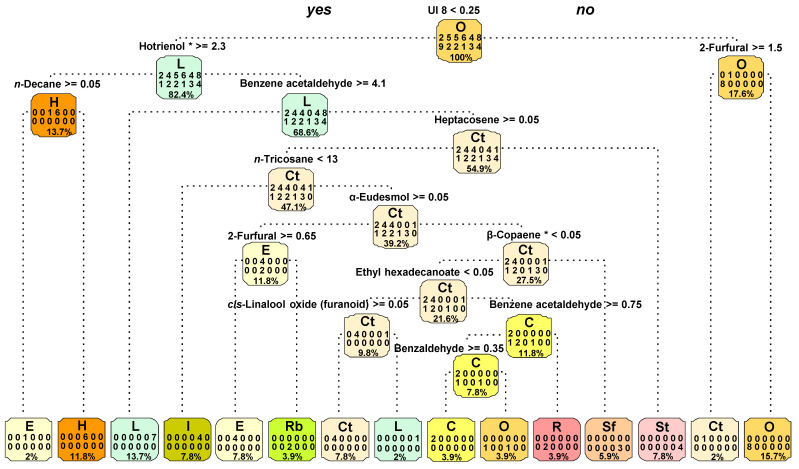
Classification tree with the honey types separated in nodes, according to volatile compounds’ discrimination. * Identification based on mass spectra only. UI: unidentified compounds; C: chestnut; Ct: carob tree; E: eucalyptus; H: bell heather; I: incense. L: lavender; O: orange; R: rape; Rb: raspberry; Sf: sunflower; St: strawberry tree. Nodes should be read as follows: The letter at the top is the node label, corresponding to the most frequent honey type in the node; the numbers represent the number of samples of each of the 12 honey types in the node (honey types in alphabetic order); and in the fourth line is the percentage of honey samples in the node (out of the 51 samples under analysis).

**Table 1 molecules-26-04970-t001:** Monofloral-labelled honeys studied, their production regions and analysed flower types, with corresponding geographical origins and volatile components found in each honey’s botanical source.

Honey Samples	Locality/Region of Honey Production	Flower’s Geographical Origin	Volatile Components Present in Selected Flowers and Simultaneously in Corresponding Honeys
Carob tree (Ct)		Estremadura	α-Pinene, *cis-* and *trans*-linalool oxide (furanoid), *cis*- and *trans*-linalool oxide (pyranoid) *, UI 8, methyl anthranilate, 1,2-dihydro-1,1,6-trimethyl-naphtalene *, α-eudesmol
Ct1	Lagos/Algarve		
Ct2	Tavira/Algarve		
Ct3	Olhão/Algarve		
Ct4	Tavira/Algarve		
Ct5	Tavira/Algarve		
Chestnut (C)			
C1	Macedo de Cavaleiros/Trás-os-Montes e Alto Douro	Trás-os-Montes e Alto Douro	Benzaldehyde, α-pinene, benzene acetaldehyde, acetophenone, *cis-* and *trans*-linalool oxide (furanoid), UI 8, 1,2-dihydro-1,1,6-trimethyl-naphtalene *, α-eudesmol, β-eudesmol
C2	Vila Pouca de Aguiar/Trás-os-Montes e Alto Douro		
Eucalyptus (E)			
E1	Ponte de Lima/Minho	Beira Litoral	α-Pinene, *cis*-linalool oxide (furanoid), aromadendrene, α-eudesmol, β-eudesmol
E2	Penamacor/Beira Baixa		
E3	Marvão/Alto Alentejo		
E4	Vila Pouca de Aguiar/Trás-os-Montes e Alto Douro		
E5	Sesimbra/Estremadura		
Bell heather (H)			
H1	Fundão/Beira Baixa	Trás-os-Montes e Alto Douro	Benzaldehyde, benzene acetaldehyde, UI 8, 1,2-dihydro-1,1,6-trimethyl-naphtalene *
H2	Boticas/Trás-os-Montes e Alto Douro		
H3	Pampilhosa da Serra/Beira Baixa		
H4	Ponte da Barca/Minho		
H5	Vila Pouca de Aguiar/Trás-os-Montes e Alto Douro		
H6	Melgaço/Minho		
Incense (I)			
I1	São Miguel/Açores	São Miguel/Açores	Benzyl salicylate *
I2	Pico/Açores		
I3	São Miguel/Açores		
I4	Pico/Açores		
Lavender (L)			
L1	Castelo Branco/Beira Baixa	Beira Baixa	No specific components found
L2	Castelo Branco/Beira Baixa		
L3	Macedo de Cavaleiros/Trás-os-Montes e Alto Douro		
L4	Mértola/Baixo Alentejo		
L5	Penamacor/Beira Baixa		
L6	Évora/Alto Alentejo		
L7	Seixal/Estremadura		
L8	Sesimbra/Estremadura		
Orange (O)			
O1	Faro/Algarve	Estremadura	α-Pinene; lilac aldehydes A *, B *, C *; UI 8; *cis*-myrcenol; indole; methyl anthranilate
O2	Tavira/Algarve		
O3	Sesimbra/Estremadura		
O4	Aljezur/Algarve		
O5	Odemira/Alentejo Litoral		
O6	Odemira/Alentejo Litoral		
O7	Lagos/Algarve		
O8	Odemira/Alentejo Litoral		
O9	Portimão/Algarve		
Rape (R)			
R1	Odemira/Alentejo Litoral	†	
R2	Odemira/Alentejo Litoral		
Raspberry (Rb)			
Rb1	Sesimbra/Estremadura	†	
Rb2	Odemira/Alentejo Litoral		
Rosemary (Ro)			
Ro1	Sesimbra/Estremadura	†	
Sunflower (Sf)			
Sf1	Odemira/Alentejo Litoral	Baixo Alentejo	α-Pinene
Sf2	Aljezur/Algarve		
Sf3	Ferreira do Alentejo/Baixo Alentejo		
Strawberry tree (St)			
St1	Aljustrel/Baixo Alentejo	Estremadura	α-Isophorone *, 4-keto-isophorone *, UI 8, 2,3,5-trimethylphenol, 3,4,5-trimethylphenol, 1,2-dihydro-1,1,6-trimethyl-naphtalene *
St2	Ferreira do Alentejo/Baixo Alentejo		
St3	Lagos/Algarve		
St4	Aljezur/Algarve		
Total	51		

* Identification based on mass spectra only; UI: unidentified compounds; † unavailable for extraction.

**Table 2 molecules-26-04970-t002:** Minimum and maximum percentage range of the main components (≥5%) isolated by hydrodistillation from the different monofloral honeys grouped according to cluster analysis. For samples grouped in each of the clusters and sub-clusters, see Figure 2. Fully detailed composition is provided in SM Appendix A.

Components	RI	Cluster I														Cluster II	
		Ia		Ib	Ic	Id	Ie	If		Ig		Ih		Ii			
		Min.	Max.					Min.	Max.	Min.	Max.	Min.	Max.	Min.	Max.	Min.	Max.
Benzene acetaldehyde	1002		9.9	8.8	3.6	0.7	0.1	0.1	0.4	1.1	3.8	0.2	3.1	17.3	24.4	0.1	0.3
*cis*-Linalool oxide (furanoid)	1045		6.5	t	0.5	t	0.1	0.1	0.3	9.0	15.1	0.2	2.9	2.9	4.3		t
*trans*-Linalool oxide (furanoid)	1059		2.4		t		t	0.1	0.1	3.9	5.1	t	1.7	1.6	2.0		t
Hotrienol *	1074		5.1	0.1	0.5	0.1	0.1			2.9	9.8		0.3	3.2	6.9		
α-Isophorone (2-cyclohexen-1-one, 3,5,5-trimethyl-)	1074		4.4			0.1	t		0.1		0.1		0.1		2.6	36.4	38.8
3,4,5-Trimethylphenol (3,4,5-hemimellitenol)	1277		1.5			t	t	t	0.1		t		0.2		0.6	4.4	13.3
Decanoic acid	1356		2.2	t		t		t	t	0.2	6.4		t	0.5	3.4	t	0.4
*n*-Heptadecane	1700	1.4	5.9	1.0	3.3	2.8	7.9	1.2	2.0	1.0	3.9	0.8	2.6	1.2	1.3	0.3	0.6
*n*-Nonadecane	1900	2.4	14.7	1.8	4.5	6.0	25.6	2.1	3.8	1.4	6.1	2.0	5.1	2.9	3.1	0.6	1.8
Hexadecanoic acid (palmitic acid)	1908		7.4	1.4	2.1	13.4	5.4	2.5	4.9		2.2	3.3	7.7	0.6	2.7	1.9	2.9
Oleyl alcohol * [(*Z*)-octadec-9-en-1-ol]	2044		0.3			11.4	0.1										
1-Octadecanol (stearyl alcohol)	2095		4.5	0.9			0.3				3.1		1.1	1.4	3.4		
Heneicosene *	2088		5.4		t	0.4	0.2	1.3	4.1		2.0		1.6			0.3	1.4
Methyl oleate (methyl *cis*-9-octadecenoate)	2096		2.1	6.3							t				t		1
*n*-Heneicosane	2100	3.3	11.4	2.4	4.3	6.8	15.1	24.0	25.5	1.3	5.0	2.7	4.0	2.3	3.1	2.9	7.0
Oleic acid (*cis*-9-octadecenoic acid)	2119		13.7	1.8	2.1	2.6		t	1.0	1.3	3.9	17.3	38.8	0.4	2.6	0.8	8.7
Linoleic acid (*cis*-9, *cis*-12-octadecadienoic acid)	2140		2.3								1.3		9.4				
Ethyl oleate	2151		10.8	1.1	1.7	2.0	0.6				0.3		9.3		2.1		
*(Z)*-9-Tricosene	2287		6.1								2.1						
*n*-Tricosane	2300	13.9	29.3	12.2	12.9	14.6	13.4	9.4	13.8	6.0	12.6	7.5	14.8	7.3	8.0	1.2	2.5
*n*-Pentacosane	2500	0.3	16.1	6.0	6.2	6.7	7.0	8.7	11.3	2.7	6.0	4.4	7.8	3.0	5.0	1.5	2.3
Heptacosene	2667		6.2	1.2	1.7	0.5	0.2	t	t	0.8	1.4	0.1	5.3	0.5	2.0		
*n*-Heptacosane	2700		8.2	2.6	2.1	2.0	2.7	4.3	5.8	1.1	2.3	t	7.1	1.4	3.4	1.0	2.6
*n*-Octacosane	2800		4.6								0.1		0.3		0.4		
**% Identification**		69.6	96.4	70.0	68.8	87.0	95.6	74.1	90.2	73.2	89.5	81.5	90.4	86.5	89.5	80.2	88.7
**Grouped components**																	
**Terpenes and derivatives**																	
Hemiterpene hydrocarbons			t	t	t						t		t	t	t		
Monoterpene hydrocarbons			1.2			0.1			0.1	t	0.7		0.5	t	0.5		
Oxygen-containing monoterpenes			12.1	0.3	1.0	0.1	0.2	1.9	2.2	19.9	27.4	1.0.	5.4	10.5	11.6	0.1	0.6
Sesquiterpene hydrocarbons			0.7	0.3	1.8	0.2	t				0.1		0.2	t	0.2		t
Oxygen-containing sesquiterpenes			1.8	0.5	t	t	t	t	t		t		0.8	t	0.2	t	4.2
Oxygen-containing diterpenes			0.4					t	t		t						
Apocarotenoids			5.2	t	t	0.1	t		0.1	0.4	0.7		0.1	0.6	3.7	41.6	44.3
**Amino acid derivatives and phenylpropanoids**																	
Benzoic acid derivatives			1.3	1.0	1.1	t	t	t	t	0.2	0.5		0.7	1.6	2.6	t	0.3
Phenylpropenes			1.0	1.1	t			t	t		1.1		0.3	t	0.5	t	t
Aromatic amino acid derivatives		0.1	9.9	8.8	3.6	1.1	0.1	1.4	2.0	1.1	3.8	0.9	3.1	17.3	25.2	7.4	18.6
**Fatty acids and derivatives**																	
Green leaf volatiles (GLVs)			0.1	t	t	t	t	t	t	t	0.1	t	t				
Fatty acids		1.0	18.6	3.5	4.2	18.0	5.4	3.5	5.1	9.0	11.4	20.6	45.3	3.7	9.6	2.8	12.1
Alkanes		40.8	74.2	27.7	37.1	43.2	76.9	55.7	65.7	15.2	35.9	24.5	42.7	22.2	27.0	11.2	14.9
Methyl-branched hydrocarbons			1.2	0.8	0.8	0.5	0.2	0.3	0.8		0.9		1.5	0.3	0.4	t	0.1
Other fatty acid derivatives		6.6	28.5	25.6	18.2	23.4	12.6	8.8	16.6	10.2	14.3	10.7	17.4	14.5	18.1	2.0	4.4
**Carbohydrate derivatives**			2.0	t	1.0	0.2	0.1	t	0.1	2.0	2.8	t	0.7	1.9	2.2	0.2	0.9
**Nitrogen-containing compounds**			t													t	0.1
**Sulphur-containing compounds**			t				t										
Others			2.3	0.4						t	3.5		t	0.4	1.2	t	1.4

RI: In-lab calculated retention index relative to C6-C31 *n*-alkanes on the DB-1 column. Min.: minimum; Max.: maximum; t: traces (<0.05%). * Identification based on mass spectra only.

## Data Availability

Not applicable.

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
