# Peer review of "Volatile Profile of Portuguese Monofloral Honeys: Significance in Botanical Origin Determination"

_molecules, 2021, doi:10.3390/molecules26164970_

Round 1

Reviewer 1 Report

The authors studied the volatile profile of Portuguese monofloral honeys, which might have importance in the botanical origin determination and in the field of fight against adulteration. They have thoroughly studied 51 samples from various geographical regions. The authors used solid phase microextraction and hydrodistillation to obtain volatiles. They have found 192 volatile compounds and summarised the results in two large tables (Tables 1. and 2.). It is not advantageous, that the identification of the compounds was based on mass spectra only in many cases and there were many unidentified compounds.

The manuscript is suitable for publication after minor revision.

The letters and numbers are not very readable in Figure 2. along the verticle axis.

I have not found Figure S1 in the Supplementary file and it is not mentioned at the end of the manuscript.

Author Response

Reviewer 1 comments:

The authors studied the volatile profile of Portuguese monofloral honeys, which might have importance in the botanical origin determination and in the field of fight against adulteration. They have thoroughly studied 51 samples from various geographical regions. The authors used solid phase microextraction and hydrodistillation to obtain volatiles. They have found 192 volatile compounds and summarised the results in two large tables (Tables 1. and 2.). It is not advantageous, that the identification of the compounds was based on mass spectra only in many cases and there were many unidentified compounds.

The manuscript is suitable for publication after minor revision.

The letters and numbers are not very readable in Figure 2. along the verticle axis.

RESPONSE # 1: A new figure with increased font size is now provided.

I have not found Figure S1 in the Supplementary file and it is not mentioned at the end of the manuscript.

RESPONSE # 2: The comment is fully right. We have provided the figure in separate in the original submission. This was now corrected.

Reviewer 2 Report

The topic of the manuscript is within the scope of the journal. Also, the research is original and well-designed. Nowadays, research about avoiding fraud in several food products is interesting and this resesarch is focused on that.

Some minor changes are required:

Table 1 should appear before Figure 1.

Please, provide a map with the studied locations. To add pictures of flowers could be increase the quality of the manuscript.

The last column of Table 1 should indicate volatile of flowers, no honey. This is the topic of the manuscript.

The reason of not using special tubes for SPME and authors used the original bottle.

How many replications were done? From the same bottle? Or 3 different lotes of the same honey?

Please, explain more Figure 2, what percentage means and the numbers “0 0 4 0 0 00 0 2 0 0 02”.

It is needed autors include a table con identification of volatile compounds: retention time…

Why authors do not use quantitative analysis? More information with concentration could be obtained, instead of %.

Author Response

Reviewer 2 comments:

The topic of the manuscript is within the scope of the journal. Also, the research is original and well-designed. Nowadays, research about avoiding fraud in several food products is interesting and this resesarch is focused on that.

Some minor changes are required:

Table 1 should appear before Figure 1.

RESPONSE # 3: Text was changed.

Please, provide a map with the studied locations. To add pictures of flowers could be increase the quality of the manuscript.

RESPONSE # 4: SF Fig. 1 (map) and SF Fig. 2 (flower's hydrodistillation) in Supplementary Material.

The last column of Table 1 should indicate volatile of flowers, no honey. This is the topic of the manuscript.

RESPONSE # 5: Text was changed for clarity.

The reason of not using special tubes for SPME and authors used the original bottle.

RESPONSE # 6: Preliminary experiments were run with SPME vials, but since the number of volatiles was in some cases very low, it was considered more adequate to use the original honey flask. In addition, this prevents loss of volatiles with handling.

How many replications were done? From the same bottle? Or 3 different lotes of the same honey?

RESPONSE # 7: Two replicates per honey flask as mentioned in the materials and methods.

Please, explain more Figure 2, what percentage means and the numbers “0 0 4 0 0 00 0 2 0 0 02”.

RESPONSE # 8: Both figure and legend were changed for clarity.

It is needed autors include a table con identification of volatile compounds: retention time…

RESPONSE # 9: This data was fully provided in the original supplementary file Tables.

Why authors do not use quantitative analysis? More information with concentration could be obtained, instead of %.

RESPONSE # 10: An absolute quantification requires the addition of an internal standard that is both chemically similar to the main honey compounds and that does not overlap with any of them. Since honeys have somewhat different compositions, this would require, always, this first study, in order to select the type of compound that better fitted as standard for each honey. Now that we could understand the composition of each honey and their eventual chemical correlation, this other study may be initiated to ascertain absolute amounts of compounds, particularly for those honey that will show relevant biological properties.